behaviour/cognition

behavioural flexibility, carrion crows, domestic chickens, motor inhibitory control, reversal learning

**Author for correspondence:**
Claudia A. F. Wascher
e-mail: claudia.wascher@gmail.com

# Learning and motor inhibitory control in crows and domestic chickens

Claudia A. F. Wascher[1], Katie Allen[1] and Georgine Szipl[2]

[1]Behavioural Ecology Research Group, School of Life Sciences, Anglia Ruskin University, Chelmsford, UK
[2]Konrad Lorenz Forschungsstelle, Core facility, University of Vienna, Gruenau, Austria

CAFW, 0000-0003-4360-363X; GS, 0000-0001-5211-8201

Cognitive abilities allow animals to navigate through complex, fluctuating environments. In the present study, we tested the performance of a captive group of eight crows, *Corvus corone* and 10 domestic chickens, *Gallus gallus domesticus*, in the cylinder task, as a test of motor inhibitory control and reversal learning as a measure of learning ability and behavioural flexibility. Four crows and nine chickens completed the cylinder task, eight crows and six chickens completed the reversal learning experiment. Crows performed better in the cylinder task compared with chickens. In the reversal learning experiment, species did not significantly differ in the number of trials until the learning criterion was reached. The performance in the reversal learning experiment did not correlate with performance in the cylinder task in chickens. Our results suggest crows to possess better motor inhibitory control compared with chickens. By contrast, learning performance in a reversal learning task did not differ between the species, indicating similar levels of behavioural flexibility. Interestingly, we describe notable individual differences in performance. We stress the importance not only to compare cognitive performance between species but also between individuals of the same species when investigating the evolution of cognitive skills.

## 1. Introduction

Cognitive abilities allow animals to navigate through complex, fluctuating environments [1] and critically affect the survival and fitness of individuals [2]. Examples of cognitive abilities include behavioural flexibility, an animal's ability to alter its behaviour in response to a novel stimulus, modify responses to familiar stimuli or inhibit previously successful behavioural strategies. A common test of behavioural flexibility is the reversal learning task, where an individual has to learn that one cue is rewarded over another in a discrimination task

(acquisition phase) and subsequently must respond to the previously unrewarded stimulus, when the cue-reward contingencies are switched in the reversal phase of the experiment [3].

Motor inhibitory control is the ability to control a response in order to choose a different course of action [4]. Environmental uncertainty can negatively affect inhibitory control. Pheasants, *Phasianus colchicus*, for whom a previously learned association between a visual cue and a food reward was perturbed to simulate environmental uncertainty, performed worse in an inhibitory control task compared with control individuals [5]. A common test of behavioural inhibition is the cylinder task. In this detour task, individuals are trained to take a food reward from an opaque cylinder or tube, which is open at both ends. Once individuals are reliably doing this, the opaque cylinder is replaced by a transparent one. The focal individual's ability to inhibit the motor impulse to try to reach the reward through the long side of the cylinder, which forms a transparent barrier between the individual and the reward, and detour to take the reward from the open ends is assessed [6]. In recent years, behavioural inhibition performance has been tested in a wide range of species (for example: cats, *Felis catus* [7]; common waxbills, *Estrilda astrild* [8]; dogs, *Canis familiaris* [9]; goats, *Capra aegagrus hircus* [10]; great tits, *Parus major* [11]; guppies, *Poecilia reticulata* [12]; sailfin molly, *Poecilia latipinna* [13]; vervet monkeys, *Chlorocebus pygerythrus* [14]). A large-scale study by MacLean *et al.* [15] compared inhibitory abilities in 36 species and found a positive correlation between absolute brain size and motor inhibitory control [15]. Among birds tested, different corvid species (common ravens, *Corvus corax*, New Caledonian crows, *Corvus moneduloides* and Western jackdaws, *Corvus monedula*) displayed the highest levels of motor inhibitory control [16]. In the past, the reversal learning task, next to a test of behavioural flexibility, also has been considered a test of behavioural inhibition, as individuals have to inhibit a previously learned behaviour to succeed when the reward contingency changes [17–19]. However, based on tests in humans, primates and rodents, this view has recently been contested [3,20–23].

In addition to species differences, cognitive performance in animals also differs between individuals, for example, based on sex [24] or personality [25]. Individual differences in cognitive performance are relatively understudied and only recently has the importance of understanding the causes and consequences of individual variation in cognitive performance been acknowledged [26]. In addition to understanding variation in cognitive performance between species, to fully understand the evolution of different cognitive abilities, it is also critical to further investigate the temporal (correlations in performance across different times) and contextual (correlations in performance across different tasks) repeatability of cognitive performance [27].

In the last decades, the cognitive abilities, including behavioural flexibility and motor inhibitory control of corvids, have been heavily studied [28,29]. For example, black-billed magpies, *Pica hudsonia*, performed similarly compared with different monkey species in a basic concept learning task [30] and jungle crows, *Corvus macrorhynchos*, learned to discriminate shapes and form concepts [31]. In a reversal learning task, New Caledonian crows, a tool-using corvid species and carrion crows, a closely related non-tool-using species, showed similar performance, suggesting that tool use is not causing enhanced learning performance in crows [32]. Corvids paralleled the performance of great apes in a motor-inhibition task (common ravens: 100%; jackdaws: 97%; New Caledonian crows: 92% success) [16].

The social system an animal belongs to is thought to be a key driver of their cognitive skills [33], therefore, species that have similar social systems to crows could be expected to have similar cognitive skills to them. Chickens, like crows, organize in complex social systems [29,34]; however, few studies have investigated their cognitive abilities and hence, chickens had less opportunities to demonstrate their cognitive abilities. In a study by Ferreira *et al.* [35] domestic chickens showed low levels of motor inhibitory control in the cylinder task (32% success) [35]. Motor inhibitory control was further affected by individual ranging habits. Individuals with a higher movement range showed a poorer performance compared with low ranging individuals. In red junglefowl, *Gallus gallus*, cognitive flexibility and exploratory behaviour correlated in age and sex-dependent manner. In a reversal learning task, it took chicks 7–64 (mean ± s.e., 24.02 ± 1.61) trials to learn to initially discriminate between two colours and adult female chickens 15–93 trials (mean ± s.e., 46.32 ± 3.75). In the reversal phase of the same study, chicks learned the reversed reward contingency after 10–62 trials (mean ± s.e., 32.98 ± 1.76) and adult female chickens learned this after 24–100 trials (mean ± s.e., 65.22 ± 4.53). More explorative chicks in this study showed higher cognitive flexibility compared with less explorative ones, while the opposite association was found for adult females [36]. Both groups, corvids and chicken organize in complex social systems, which is assumed a driving factor in the evolution of cognitive skills.

…widely considered 'brainy' birds [28], thus we could expect high levels of motor-inhibition, and faster reversal learning in crows, compared with chickens. However, as recent research challenges this view and highlights the

cognitive abilities of chickens [37], we do not necessarily share this expectation. Instead, we aim to contribute to knowledge about individual performances in cognitive tasks in two well-established model species.

In the present study, we investigated the performance of crows and domestic chickens in two cognitive tasks, the cylinder task as a test of motor inhibitory control and reversal learning as a measure of learning ability and behavioural flexibility. By contrast to domestic chickens, crows are widely considered 'brainy' birds [28], thus we could expect high levels of motor-inhibition, and faster reversal learning in crows, compared with chickens. However, as recent research challenges this view and highlights the cognitive abilities of chickens [37], we do not necessarily share this expectation. Instead, we aim to contribute to knowledge about individual performances in cognitive tasks in two well-established model species. With our study, we intend to add to existing data on the performances of different species in standardized experimental paradigms, which should be relatively comparable to each other. Applying two widely used experimental test paradigms in which both species, at least physically, can perform without difficulty, i.e. removing a reward from a tube and lids to access food, allows a reasonably fair comparison of cognitive performance between these two species.

# 2. Methods

## 2.1. Subjects and housing

The total study population consisted of eight crows, seven carrion crows, *Corvus corone corone* and one hooded crow, *Corvus corone cornix*. All crows were tested at the Konrad Lorenz research station (KLF), Grünau, Austria from October 2010 to August 2012. Crows were held in large outdoor aviaries (15 m$^2$), in two male–female pairs, one trio (one male, two females) and one single housed individual. Testing was conducted either in the main aviary with one animal visually and spatially separated from the others or in a separate, spacious testing room, which birds entered voluntarily upon request. Rewards consisted of greaves, i.e. fried pork skin and cheese. All focal individuals were hand-raised between 2007 and 2011, either at the KLF or by private people. Crows were captured and brought into captivity by private individuals when ejected from the nest at a young age by unfavourable weather conditions.

Domestic chickens, *Gallus gallus domesticus*, were housed at a stable yard in Piddinghoe, Newhavenin East Sussex and tested from June to August 2019. The chickens' outdoor ranging area was 334 m$^2$; including the indoor area they were kept in at night, which was 13 m$^2$. The flock consisted of 13 female individuals and one male, hatched in 2017 or 2018. The chickens had outdoor and indoor access during the day and were kept in a shed at night. Ten female chickens were selected based on their tameness and willingness to follow the experimenter into a spacious test enclosure, visually and spatially separated from all other individuals. Of these 10, one individual was excluded from the experiment due to becoming agitated and reluctant to enter the test area in the course of the experiment. Chickens were rewarded with grapes.

All procedures were conducted in accordance with the ASAB/ABS guidelines for the treatment of animals in behavioural research. Experiments complied with Austrian and UK government guidelines. As experiments were entirely non-invasive, no further animal experimental license was required. Experiments on domestic chickens have been approved by the School of Life Sciences ethics panel, Anglia Ruskin University. All individuals participated and entered the experimental compartments voluntarily. Individuals were held in captivity before and after the completion of the present study.

## 2.2. Experiment 1: the detour task

Four crows, two males and two females and nine chickens were tested in this task. Due to time constraints, not all individuals could be tested. In the first stage of the task, the experimenter placed a food reward in an opaque cylinder (approx. 30 cm in length and 10 cm in diameter), while the focal individual was watching, so the bird see how to get the reward. The cylinder needed to be sufficiently long that the birds needed to insert their head to obtain the reward at the centre of the cylinder, but not too large so that the birds could not enter the cylinder with their whole body. Crows received 10 trials per session; chickens received five trials per session, to keep the motivation of individuals high. A correct response was when an individual retrieved the reward from either of the openings, without any prior contact with the long side of the cylinder. To pass the first stage of the detour task, individuals were required to complete at least five successful retrieves in a row. Once the

first stage of the experiment was completed, individuals were presented with the transparent cylinder of the same dimensions compared with the opaque cylinder. Similar to the first stage, a food reward was placed in the cylinder in front of the focal individual. Crows received one test session consisting of 10 trials and chickens two test sessions consisting of five trials, because chickens were less motived to complete a high number of trials. Therefore, overall, each individual received 10 trials. The number of correct responses, i.e. individuals reaching directly to the reward without touching the long side of the cylinder first, was recorded similarly to stage one of the experiments. Individuals able to inhibit trying to reach the reward directly from the front but retrieving the reward from the long side of the cylinder, and thus being able to detour, are considered better at motor-inhibition.

## 2.3. Experiment 2: the reversal task

Eight crows, three males and five females and nine chickens participated in this experiment. Data on learning performance from four focal crows have been previously published [32]. The apparatus consisted of two feeders that were mounted 30 cm apart on a wooden board. Feeders could be covered with paper lids of different colours (orange and blue). Prior to the test, a habituation period was conducted, familiarizing the focal individuals with the test apparatus and procedure. First, individuals were habituated to take rewards from the feeders. Afterwards, feeders were partially covered with a white lid, training individuals to remove the lid to retrieve the reward. In each trial, a reward was placed in one of the feeders behind a barrier and out of view of the focal animal. The lids were placed on the feeders and the apparatus positioned in front of the focal individual, who was then allowed to remove one of the two lids. The experiment consisted of two phases: an initial acquisition phase and a reversal phase. In the acquisition phase, one colour was the rewarded (S+) stimulus; half of the birds started with orange as S+ the other half with blue as S+. The success criterion for the acquisition phase was that 80% of 10 trials were correct over two consecutive sessions or 100% of 10 trials correct in one session. Once a subject met the criterion, the colour-reward contingency was reversed in the reversal phase. In the acquisition phase, subjects were given a maximum of 140 trials to reach the learning criterion, which one domestic chicken did not reach and consecutively was not included in further testing in the reversal phase of the experiment. Tests of two further chickens and one crow could not be completed in the reversal phase due to time constraints. If a subject developed a positional bias (side bias), that is, when it chose one side in six consecutive trials, a side bias correction procedure was applied until that subject chose the non-preferred side once, whereupon we reverted to the normal pseudo-randomized trial schedule and always rewarded the non-preferred side until that subject chose the non-preferred side once. Test sessions consisted of 10 trials each. The rewarded side was pseudo-randomized, with each side being rewarded 5 times per session for the crows. The maximum number of rewarded trials on one side was seven and the minimum of times a side was rewarded in one session was three in the chickens.

## 2.4. Statistical analysis

Statistical analysis was performed in R v. 3.5.3 (The R Foundation for Statistical Computing, Vienna, Austria, http://www.r-project.org). Exact Wilcoxon–Mann–Whitney tests were computed to compare test performances between crows and chickens. Spearman rank correlations were calculated using the package Hmisc [38]. Spearman rank correlations were used to investigate correlations in test performance within the reversal learning task, i.e. between acquisition and reversal phase and between tasks, i.e. between reversal learning task and tube task. Additionally, we calculated delta values in the reversal learning experiment, i.e. trials required in the reversal phase subtracted from trials until learning criterion is reached in the acquisition phase, proving an estimate of behavioural flexibility. We correlated delta values in the reversal learning experiment and performance in the cylinder task. All datasets and the R scripts used to conduct the statistical analyses are available as electronic supplementary material files.

# 3. Results

Individual performance in the cylinder task ranged from 30% to 100% correct trials in crows (mean ± s.d.: 73 ± 31) and 10 to 40% correct trials in domestic chickens (mean ± s.d.: 23 ± 12). Crows performed better

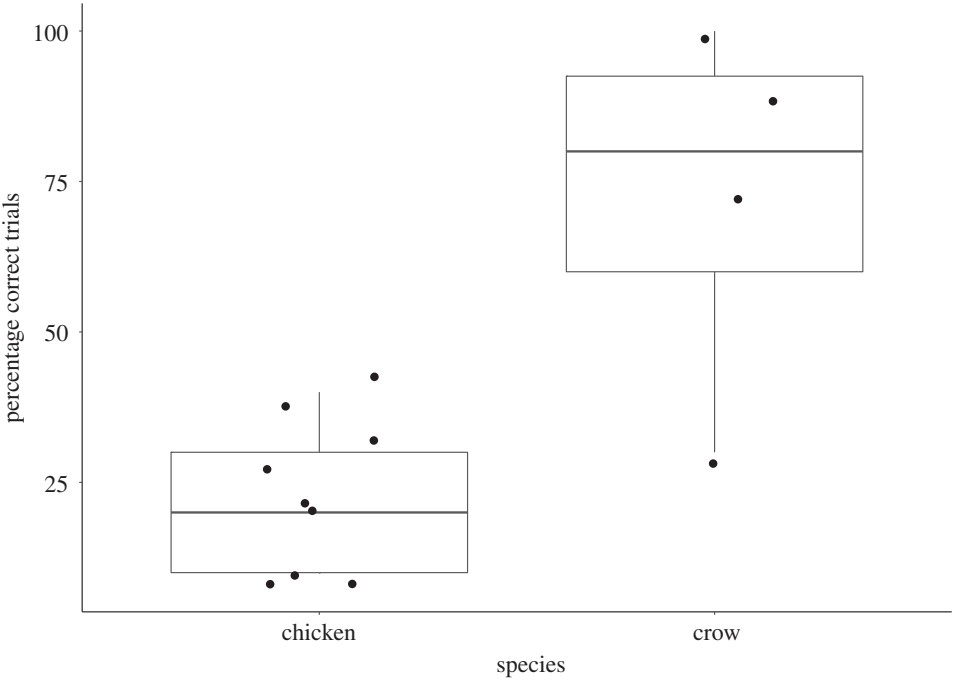

**Figure 1.** Percentage of correct test trials in the cylinder task. Box plots show the median and the interquartile range from the 25th to the 75th percentiles, black circles represent individual performance.

in the cylinder task compared with domestic chickens (Wilcoxon–Mann–Whitney test: $W = 3$, $n_{\text{chickens}} = 9$, $n_{\text{crows}} = 4$, $p = 0.023$; figure 1).

In the acquisition phase of the reversal task, individual performance in crows ranged from 31 to 118 trials until learning criterion was reached (mean ± s.d.: 73 ± 33) and in domestic chickens from 30 to 140 trials (mean ± s.d.: 71 ± 39). In the reversal phase, crows' performance ranged from 56 to 200 trials until learning criterion was reached (mean ± s.d.: 122 ± 51) and domestic chickens' performance ranged from 30 to 100 trials (mean ± s.d.: 87 ± 24). It should be noted that three chickens did not complete the reversal phase of the experiment (table 1). In the reversal learning experiment, species did not significantly differ in the number of trials until learning criterion was reached (Wilcoxon–Mann–Whitney test: acquisition phase: $W = 34.5$, $n_{\text{chickens}} = 9$, $n_{\text{crows}} = 8$, $p = 0.923$; reversal phase: $W = 12$, $n_{\text{chickens}} = 6$, $n_{\text{crows}} = 7$, $p = 0.221$).

In crows, individuals who needed fewer trials to reach learning criterion in the acquisition phase also needed fewer trials to reach criterion in the reversal phase (Spearman correlation: $r = 0.86$, $n = 7$, $p = 0.013$; figure 2). This relationship was lacking in domestic chickens (Spearman correlation: $r = -0.14$, $n = 6$, $p = 0.796$; figure 2). Performance in the learning task did not correlate with performance in the cylinder task in domestic chickens (Spearman correlation: acquisition phase: $r = 0.08$, $n = 9$, $p = 0.833$; reversal phase: $r = 0.48$, $n = 6$, $p = 0.329$; delta values: $r = -0.13$, $n = 6$, $p = 0.799$; figure 3a). As we have only tested four crows in the cylinder task, we could not calculate a correlation between learning performance and performance in the cylinder task, but we graphically illustrated results in figure 3b.

## 4. Discussion

In the present study, we describe the performance of crows and domestic chickens in a motor-inhibition task (cylinder task) and a reversal learning task. Crows performed better in the cylinder task compared with domestic chickens.

In both tasks, we describe pronounced individual variation in performance in both species. Individual variation in cognitive performance only recently came into the focus of comparative cognition research [26,47]. Presently, we can only speculate about the causes of individual differences in our sample. Individual differences in learning performance can also be caused by physical characteristics of the environment [48], social factors [49] or differences in personality [25]. They can be caused by differences in attention and motivation [50]. A recent study found individual differences in cognitive performance in Australian magpies, *Cracticus tibicen dorsalis*, to be linked to group size, with individuals in larger groups performing better in cognitive tasks [51]. Ultimately, individual

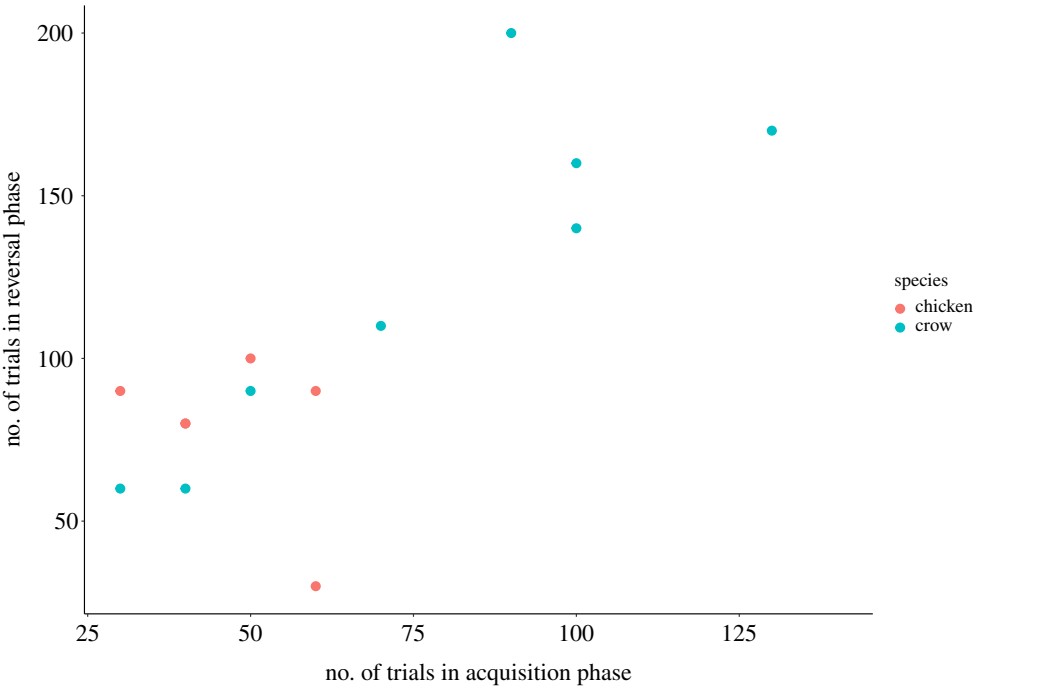

**Figure 2.** Relationship between learning performance in the acquisition phase (*x*-axis) and the reversal phase (*y*-axis) in crows (blue circles) and domestic chickens (red circles).

**Table 1.** Individuals tested in the present study, the species (crow or chicken), sex (male or female), year of hatching and previous experience in cognitive testing. Number of trials until learning criterion was reached in the acquisition and reversal phase of the learning experiment as well as percentage of trials correct in the cylinder task. Missing values indicates that the respective individual had not completed the task (reversal task: three chickens; cylinder task four crows).

| name | species | sex | hatching | experience | acquisition | reversal | cylinder |
|------|---------|-----|----------|------------|-------------|----------|----------|
| Baerchen | crow | male | 2008 | yes [32,39–46] | 31 | 62 | 100 |
| Gabi | crow | female | 2007 | yes [32,40–46] | 50 | 90 | |
| Peter | crow | female | 2007 | yes [32,39–46] | 65 | 107 | 90 |
| Toeffel | crow | female | 2008 | yes [32,39–41,45,46] | 118 | 160 | |
| Klaus | crow | male | 2009 | yes [32,39,40,44,46] | 33 | 56 | 70 |
| Resa | crow | female | 2009 | yes [32,40,42,44,46] | 102 | | 30 |
| Nino | crow | female | 2011 | yes [42] | 90 | 200 | |
| Walter | crow | male | 2011 | yes [32] | 95 | 159 | |
| BG | chicken | female | 2017/2018 | no | 100 | | 10 |
| DG | chicken | female | 2017/2018 | no | 40 | 80 | 30 |
| LB | chicken | female | 2017/2018 | no | 30 | 90 | 20 |
| N | chicken | female | 2017/2018 | no | 60 | 30 | 20 |
| O | chicken | female | 2017/2018 | no | 140 | | 40 |
| P | chicken | female | 2017/2018 | no | 60 | 90 | 40 |
| WH | chicken | female | 2017/2018 | no | 120 | | 10 |
| W | chicken | female | 2017/2018 | no | 40 | 80 | 10 |
| Y | chicken | female | 2017/2018 | no | 50 | 100 | 30 |

differences in cognitive performance can affect individuals' survival and reproductive success [26,47,52]. In humans, different psychometric tests assessing different cognitive processes are well-established; however, similar tests have only been developed and validated in a small number of non-human animal species [53]. Determining which factors underlie cognitive variation, how cognitive variation

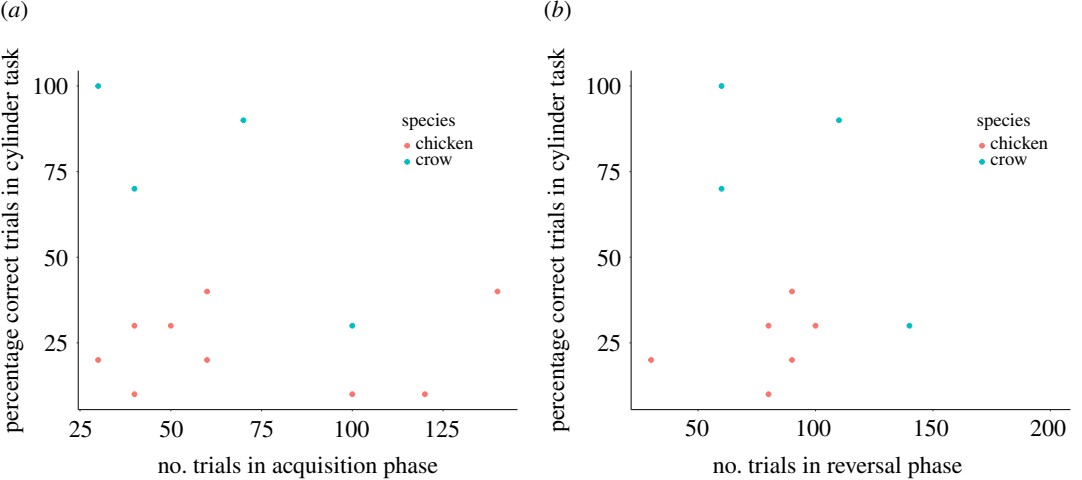

**Figure 3.** Relationship between learning performance in the acquisition phase (*a*) and reversal phase (*b*) on the *x*-axis and performance in the cylinder task (*y*-axis) in crows (blue circles) and domestic chickens (red circles).

affects individual outcomes and developing and validating tests of cognitive processes for multiple species could help to enhance our understanding of the evolution of cognitive processes.

One critical limitation of our study is the small sample size. We only tested four crows in the cylinder task, which makes statistical comparisons between but also within, the tested species difficult. Small sample sizes are not uncommon in comparative cognition. In the field of physical cognition, all studies conducted on corvids had a sample size smaller than 10 [54]. This can be explained by various factors, for example, the limited availability of captive individuals for testing, the high training effort, and connected to this, large time investment to conduct cognitive tests. Opportunities to overcome these limitations could be multi-lab collaborative studies [55,56] or meta-analyses [25]. The present study, despite the small sample size, will contribute to the growing body of literature about learning abilities in different species, and can be incorporated into larger-scale comparisons.

We must also acknowledge that our study animals may have differed in experience, which could have influenced their performance in our cognitive tests. For example regarding development and raising, social and physical environment, differences in experience with cognitive tests, and potentially, transparent materials could account for differences in cognitive performance between species. In general, it is challenging to keep groups of animals under the same conditions and experiences might even vary between individuals of the same group. Cognitive studies that use voluntary participation, as we did, may not sample the full range of personalities in a population and results might be affected by a selection bias, with 'braver' individuals participating, whereas 'shyer' individuals, especially in the case of our chickens might not have been tested [57]. Webster & Rutz [58] recently, provided a framework to allow researchers to judge potential sampling biases in their study populations [58]. They suggest social background of individuals, e.g. dominance status, self-selection, e.g. participation of bold individuals, the rearing history of animals, acclimation and habituation, natural changes in responsiveness and genetic make-up as potential factors affecting the behaviour of study animals and thus who participates in voluntary cognitive testing. We attempt to fully declare and acknowledge the 'STRANGEness' [58] of our study subjects and we discuss the potential effects of sampling bias on our results. Although our sample might have been biased, we would like to highlight that we describe pronounced individual differences within each species of our sample.

Cognition research in itself is biased and traditionally focuses on comparisons between a small group of species considered as 'cognitively complex' [59]. Our results illustrate that certain cognitive abilities, e.g. learning abilities, might not differ between species considered as cognitively complex, i.e. crows, compared with species not considered to possess cognitively advanced skills, i.e. domestic chicken, however, pronounced individual differences might exist within species.

Our results suggest corvids to perform well and chickens to perform poorly in the cylinder task, which is in line with previous findings [16,35]. This could indicate better motor inhibitory control in crows compared with chickens. It could be argued that higher motor inhibitory control in corvids compared with chickens allows them to more selectively attend and suppress pre-potent motor impulses and therefore, crows compared with chickens could be expected to perform better in tasks requiring complex decision-making processes, such as delay of gratification [41,42], inequity aversion

[44], tactical deception [60], mental time travel [61] and causal reasoning [62]. Carrion crows are a food caching species, which regularly hide food for short periods of time [63]. The regular practice of leaving food behind for later consumption could lead to better motor inhibitory control in crows compared with chickens, which are a non-caching species. Another factor which could potentially contribute to performance in the cylinder task is object permanence, i.e. an understanding that an object exists even when it is out of sight. Carrion crows have been shown to possess full Piagetian object permanence [64], whereas this cognitive ability has not been fully studied in chickens [37]. This means a better sense of object permanence in crows compared with chickens could lead to a better understanding of the reward still being present although out of sight in the opaque stage of the cylinder task, which could affect overall performance in the cylinder task. However, it also needs to be noted that chickens in previous tasks could learn to solve a detour task [65] and that poor performance can be explained by perceptual and motivational factors [66]. In summary, results of the cylinder task suggest crows to perform better compared with chickens, but further research on the underlying causes of this is desirable.

In the reversal learning task, crows and domestic chickens did not differ in performance, measured in the number of trials until learning criterion was reached, neither in the acquisition phase, nor the reversal phase of the experiment, although it should be noted that three chickens and one crow did not reach learning criterion. These results are in contrast to a previous study, in which red-billed blue magpies, *Urocissa erythroryncha*, outperformed white leghorn chickens in a serial reversal task [67], but they highlight the importance of replicating results of previous studies and not drawing final conclusions about cognitive abilities of species from a limited sample of individuals. It should be noted that the reversal learning task also requires a certain level of behavioural inhibition [68]. In the chickens, we conducted the cylinder task before the reversal learning experiment, thus previous experience with the motor-inhibition task could have affected the performance of chickens in the reversal learning task.

Next to individual variation in cognitive performance, another factor to consider when comparing cognitive performance between species is the question of which cognitive abilities different tests assess. One main advantage of the cylinder task is that it is easy and quick to conduct. The task requires low levels of habituation and training and can be successfully applied in a wide variety of species and thus has been widely applied to assess motor inhibitory control [6,15]. However, the task is not uncontroversial and several authors question its suitability to assess motor inhibition [69,70]. We can exclude that domestic chickens had prior experience with transparent materials, however, crows might have had prior experience, for example with transparent plastic bottles, provided as enrichment to the captive crows. Experience with transparent materials could be a potential cause for crows performing better compared with chickens, however, in great tits, *P. major*, general experience with a transparent wall did not improve performance in the cylinder task [11].

We found a positive correlation between learning performance in the acquisition and reversal phase in crows but not in chickens. This indicates a relationship between reward-based stimulus association learning and behavioural flexibility. Several studies evidence a lack of correlation in performance between the acquisition and reversal learning phase [71]. The described species difference between crows and chickens is interesting and warrants further investigation, however, due to the low sample size of the present study should not be over-interpreted.

In conclusion, in our experiments, crows seem to possess better motor inhibitory control compared with domestic chickens. By contrast, learning performance in a reversal learning task did not differ between crows and domestic chickens. The sample size of our study is too small to draw final conclusions about differences or similarities in cognitive performance between the species, however, we found notable individual differences in performance in both species. Our results highlight the importance for animal cognition research to move away from focusing on animals that are already presumed smart and instead broaden the range of species investigated.

Ethics. All procedures were conducted in accordance with the ASAB/ABS guidelines for the treatment of animals in behavioural research. Experiments complied with Austrian and UK government guidelines. As experiments were entirely non-invasive, no further animal experimental license was required. Experiments on domestic chickens have been approved by the School of Life Sciences ethics panel, Anglia Ruskin University. All individuals participated and entered the experimental compartments voluntarily. Individuals were held in captivity before and after completion of the present study.
Data accessibility. The data are provided in the electronic supplementary material [72].
Authors' contributions. C.A.F.W., G.S.: conceptualization; C.A.F.W., K.A. and G.S.: data curation; C.A.F.W.: formal analysis; C.A.F.W., K.A. and G.S.: writing—original draft, writing—review and editing.

All authors gave final approval for publication and agreed to be held accountable for the work performed therein.

Competing interest. At the time of writing, C.A.F.W. was a Board Member of Royal Society Open Science, but had no involvement in the review or assessment of the paper. The authors declare no conflict of interest.

Funding. We received no funding for this study.

Acknowledgements. We thank Christian Schloegl for assistance with the experimental procedure while conducting the cylinder task and two anonymous referees for valuable comments on the manuscript.

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
