## [Peer Review File · Royal Society Open Science]

Review History

RSOS-210504.R0 (Original submission)

Review form: Reviewer 1

Is the manuscript scientifically sound in its present form?

No

Are the interpretations and conclusions justified by the results?

Yes

Is the language acceptable?

Yes

Do you have any ethical concerns with this paper?

No

Have you any concerns about statistical analyses in this paper?

No

Recommendation?

Major revision is needed (please make suggestions in comments)

Comments to the Author(s)

Please see attached file (Appendix A).

Review form: Reviewer 2 (Mark O'Hara)**Is the manuscript scientifically sound in its present form?**

Yes

Are the interpretations and conclusions justified by the results?

Yes

Is the language acceptable?

Yes

Do you have any ethical concerns with this paper?

No

Have you any concerns about statistical analyses in this paper?

No

Recommendation?

Major revision is needed (please make suggestions in comments)

Comments to the Author(s)

This study by Wascher, Allen and Szpl shows how crows and domestic chickens differ in their motor inhibition skills, but not necessarily in overall behavioural flexibility.

To this end, the authors provided a limited sample of individuals of both species with a) a single reversal-learning task investigating behavioural flexibility and b) a cylinder task to test for inhibitory control.

Finally, the authors highlight the importance of individual variation in performance in both species.

I have two main content remarks regarding this study (and try to provide suggestions on how to address these):

I was surprised to see that the authors did not elaborate more on the conceptual overlaps and implications of choosing these two tasks:

Reversal learning (and especially the reversal phase) has been suggested in older literature to comprise of the ability to inhibit responses to stimuli with a previous positive valence (being rewarded) and the ability to shift the attention to previously unrewarded stimuli (e.g. Dias et al., 1996; Lai et al., 1995; Macphail, 1971, 1976).

However, more recent studies (Powers 1989; Bonte, Flemming & Fagot 2011; Fagot, Bonte, & Hopkins 2013; Bonte, Kemp & Fagot 2014) agree that reversal learning rather measures the ability to shift the attention (as a measure for flexibility), than the inhibition of responding to a previously rewarded stimuli. It seems that what the authors find is along these lines of this newer literature, as they did not find a correlation between the reversal task and the inhibition task.

Nevertheless, it might be telling to compare delta values in the reversal task (trials required in the reversal subtracted by trials in the acquisition). If there is a correlation between pure inhibition (as in the cylinder task) and such values (which would account for general differences in learning) would indicate that inhibition plays a more prominent role in the reversal phase. Whereas, if there is still no correlation between tasks, it would strengthen the argument that reversal learning instead measures the ability to shift attention. However, I understand this analysis might be challenging given the low sample size of individuals performing in all stages (acquisition and reversal) and both tasks (reversal learning and cylinder task).

As the authors highlight the importance of considerable individual variation in performance, I would have been interested in an analysis and statement as to what factors (sex, age, personalities?) might influence this variation. Possibly a model on the primary principle component of the measures could inform if any specific factor would affect performance?

Below are some minor stylistic comments:

Line 13: should read "the ability of an animal to alter its behaviour" or "the ability of animals to alter their behaviour"

Lines 19-20: It would be clearer just to state how many individuals complete the entire reversal-learning task

Line 37: should read "an animal's ability to alter its behaviour"

Line 90: remove the opening parenthesis before Kabadayi et al.

Line 129: missing "The" before flock

Line 233: should read: "...pronounced individual differences within..."

Line 234: check the spelling of "focuses"

Line 238: should read: "individual differences might exist..."

Line 254: should read: "...crows compared to chickens."

Line 257: check the spelling of "suppress"

Line 275: should read: "...the question of which cognitive abilities..."

Please also check the correct use of commas in several places.

Decision letter (RSOS-210504.R0)

Dear Dr Wascher

The Editors assigned to your paper RSOS-210504 "Learning and motor-impulse control in crows and domestic chickens" have now received comments from reviewers and would like you to revise the paper in accordance with the reviewer comments and any comments from the Editors. Please note this decision does not guarantee eventual acceptance.

Please submit your revised manuscript and required files (see below) no later than 21 days from today's (ie 02-Aug-2021) date. Note: the ScholarOne system will 'lock' if submission of the revision is attempted 21 or more days after the deadline. If you do not think you will be able to meet this deadline please contact the editorial office immediately.

on behalf of Professor Kevin Padian (Subject Editor)
openscience@royalsociety.org

Associate Editor Comments to Author:

Many thanks for your patience while we sought reviewers of the paper - this has been more difficult than we'd hoped, but the reviewers who have kindly reported on the paper recommend a number of changes that we'd like you to incorporate into the manuscript. If you need a short extension on the revision deadline, please let the editorial office know.

Reviewer comments to Author:

Reviewer: 1
Comments to the Author(s)
Please see attached file.

Reviewer: 2

Comments to the Author(s)

This study by Wascher, Allen and Szpl shows how crows and domestic chickens differ in their motor inhibition skills, but not necessarily in overall behavioural flexibility.

To this end, the authors provided a limited sample of individuals of both species with a) a single reversal-learning task investigating behavioural flexibility and b) a cylinder task to test for inhibitory control.

Finally, the authors highlight the importance of individual variation in performance in both species.

I have two main content remarks regarding this study (and try to provide suggestions on how to address these):

I was surprised to see that the authors did not elaborate more on the conceptual overlaps and implications of choosing these two tasks:

Reversal learning (and especially the reversal phase) has been suggested in older literature to comprise of the ability to inhibit responses to stimuli with a previous positive valence (being rewarded) and the ability to shift the attention to previously unrewarded stimuli (e.g. Dias et al., 1996; Lai et al., 1995; Macphail, 1971, 1976).

However, more recent studies (Powers 1989; Bonte, Flemming & Fagot 2011; Fagot, Bonte, & Hopkins 2013; Bonte, Kemp & Fagot 2014) agree that reversal learning rather measures the ability to shift the attention (as a measure for flexibility), than the inhibition of responding to a previously rewarded stimuli. It seems that what the authors find is along these lines of this newer literature, as they did not find a correlation between the reversal task and the inhibition task.

Nevertheless, it might be telling to compare delta values in the reversal task (trials required in the reversal subtracted by trials in the acquisition). If there is a correlation between pure inhibition (as in the cylinder task) and such values (which would account for general differences in learning) would indicate that inhibition plays a more prominent role in the reversal phase.

Whereas, if there is still no correlation between tasks, it would strengthen the argument that reversal learning instead measures the ability to shift attention.

However, I understand this analysis might be challenging given the low sample size of individuals performing in all stages (acquisition and reversal) and both tasks (reversal learning and cylinder task).

As the authors highlight the importance of considerable individual variation in performance, I would have been interested in an analysis and statement as to what factors (sex, age, personalities?) might influence this variation. Possibly a model on the primary principle component of the measures could inform if any specific factor would affect performance?

Below are some minor stylistic comments:

Line 13: should read "the ability of an animal to alter its behaviour" or "the ability of animals to alter their behaviour"

Lines 19-20: It would be clearer just to state how many individuals complete the entire reversal-learning task

Line 37: should read "an animal's ability to alter its behaviour"

Line 90: remove the opening parenthesis before Kabadayi et al.

Line 129: missing "The" before flock

Line 233: should read: "...pronounced individual differences within..."

Line 234: check the spelling of "focuses"

Line 238: should read: "individual differences might exist..."

Line 254: should read: "...crows compared to chickens."

Line 257: check the spelling of "suppress"

Line 275: should read: "...the question of which cognitive abilities..."

Please also check the correct use of commas in several places.

===PREPARING YOUR MANUSCRIPT===

===PREPARING YOUR REVISION IN SCHOLARONE===

Author's Response to Decision Letter for (RSOS-210504.R0)

See Appendix B.

RSOS-210504.R1 (Revision)

Review form: Reviewer 1

Is the manuscript scientifically sound in its present form?

Yes

Are the interpretations and conclusions justified by the results?

Yes

Is the language acceptable?

Yes

Do you have any ethical concerns with this paper?

No

Have you any concerns about statistical analyses in this paper?

No

Recommendation?

Accept with minor revision (please list in comments)

Comments to the Author(s)

Please see attached file (Appendix C).

Review form: Reviewer 2

Is the manuscript scientifically sound in its present form?

Yes

Are the interpretations and conclusions justified by the results?

Yes

Is the language acceptable?

Yes

Do you have any ethical concerns with this paper?

No

Have you any concerns about statistical analyses in this paper?

No

Recommendation?

Accept with minor revision (please list in comments)

Comments to the Author(s)

The revised manuscript by Wascher, Allen and Szpil: 'Learning and motor inhibitory control in crows and domestic chicken' has largely improved from the previous version I reviewed.

I'm happy to see how diligently the authors have taken all comments into consideration.

Based on the small sample size the generalisability of the results is of course limited. However, the authors fully acknowledge this and highlight the importance of individual factors contributing to the results.

In this sense it adds to a body of literature and a current trend to consider a subjects 'STRANGENESS' when investigating behavioural/cognitive functions.

I concur that ultimately, one forte of this study will be an additive contribution to a larger scale comparison with regard to individual differences. To this end I think it would be beneficial to provide as many details as possible for all subjects tested.

I have seen that age was listed in the ESM, but I wonder if it was possible to also provide information for each individual on other factors that are discussed to influence performance (e.g. housing conditions, rearing conditions, experimental experience, etc.) within the data table.

Decision letter (RSOS-210504.R1)

Dear Dr Wascher

On behalf of the Editors, we are pleased to inform you that your Manuscript RSOS-210504.R1 "Learning and motor-impulse control in crows and domestic chickens" has been accepted for publication in Royal Society Open Science subject to minor revision in accordance with the referees' reports. Please find the referees' comments along with any feedback from the Editors below my signature.

Please submit your revised manuscript and required files (see below) no later than 7 days from today's (ie 20-Sep-2021) date. Note: the ScholarOne system will 'lock' if submission of the revision is attempted 7 or more days after the deadline. If you do not think you will be able to meet this deadline please contact the editorial office immediately.

Please note article processing charges apply to papers accepted for publication in Royal Society Open Science (<https://royalsocietypublishing.org/rsos/charges>). Charges will also apply to papers transferred to the journal from other Royal Society Publishing journals, as well as papers submitted as part of our collaboration with the Royal Society of Chemistry

(<https://royalsocietypublishing.org/rsos/chemistry>). Fee waivers are available but must be requested when you submit your revision (<https://royalsocietypublishing.org/rsos/waivers>).

on behalf of Kevin Padian (Subject Editor)
openscience@royalsociety.org

Associate Editor Comments to Author:

Comments to the Author:

The reviewers have suggested a few minor modifications or clarifications, but the work is well on the way to being ready for acceptance - congratulations and we'll look forward to receiving a final version that incorporates these modifications.

Reviewer comments to Author:

Reviewer: 1

Comments to the Author(s)

Please see attached file

Reviewer: 2

Comments to the Author(s)

The revised manuscript by Wascher, Allen and Szpił: 'Learning and motor inhibitory control in crows and domestic chicken' has largely improved from the previous version I reviewed.

I'm happy to see how diligently the authors have taken all comments into consideration.

Based on the small sample size the generalisability of the results is of course limited. However, the authors fully acknowledge this and highlight the importance of individual factors contributing to the results.

In this sense it adds to a body of literature and a current trend to consider a subjects 'STRANGENESS' when investigating behavioural/cognitive functions.

I concur that ultimately, one forte of this study will be an additive contribution to a larger scale comparison with regard to individual differences. To this end I think it would be beneficial to provide as many details as possible for all subjects tested.

I have seen that age was listed in the ESM, but I wonder if it was possible to also provide information for each individual on other factors that are discussed to influence performance (e.g. housing conditions, rearing conditions, experimental experience, etc.) within the data table.

===PREPARING YOUR MANUSCRIPT===

===PREPARING YOUR REVISION IN SCHOLARONE===

- Any electronic supplementary material (ESM).
- If you are requesting a discretionary waiver for the article processing charge, the waiver form must be included at this step.
- If you are providing image files for potential cover images, please upload these at this step, and inform the editorial office you have done so. You must hold the copyright to any image provided.
- A copy of your point-by-point response to referees and Editors. This will expedite the preparation of your proof.

- Ensure that your data access statement meets the requirements at <https://royalsociety.org/journals/authors/author-guidelines/#data>. You should ensure that you cite the dataset in your reference list. If you have deposited data etc in the Dryad repository, please only include the 'For publication' link at this stage. You should remove the 'For review' link.
- If you are requesting an article processing charge waiver, you must select the relevant waiver option (if requesting a discretionary waiver, the form should have been uploaded at Step 3 'File upload' above).
- If you have uploaded ESM files, please ensure you follow the guidance at <https://royalsociety.org/journals/authors/author-guidelines/#supplementary-material> to include a suitable title and informative caption. An example of appropriate titling and captioning may be found at https://figshare.com/articles/Table_S2_from_Is_there_a_trade-off_between_peak_performance_and_performance_breadth_across_temperatures_for_aerobic_scope_in_teleost_fishes_/3843624.

Author's Response to Decision Letter for (RSOS-210504.R1)

See Appendix D.

Decision letter (RSOS-210504.R2)

Dear Dr Wascher,

I am pleased to inform you that your manuscript entitled "Learning and motor inhibitory control in crows and domestic chickens" is now accepted for publication in Royal Society Open Science.

Please ensure that you send to the editorial office an editable version of your accepted manuscript, and individual files for each figure and table included in your manuscript. You can send these in a zip folder if more convenient. Failure to provide these files may delay the

processing of your proof. You may disregard this request if you have already provided these files to the editorial office.

on behalf of Kevin Padian (Subject Editor)
openscience@royalsociety.org

Appendix A

REVIEW FOR LEARNING AND MOTOR-IMPULSE CONTROL IN CROWS AND DOMESTIC CHICKENS

Overview of study

This study aimed to investigate inhibitory control and behavioural flexibility in crows and domestic fowl. Inhibitory control was measured with a detour task and behavioural flexibility was measured in a reversal learning test. Crows performed better in the inhibitory control test than chickens and no difference was found in terms of reversal learning. The authors suggest that this implies crows are better at inhibitory control than chickens, but also point out that their small sample sizes make statistical comparison difficult. The study also touched upon within species differences/individual differences in cognition and repeatability of cognitive performance across different tests.

I reviewed an earlier version of this manuscript and I was excited to see what it looked like now. I feel that, overall, it is much improved, and I am grateful that you seem to have taken my previous comments into consideration. I see promise in this paper to be interesting in itself and a good addition to the field of animal cognition – though it is not quite there yet.

I do have some points that I would like to see addressed before I would consider this study acceptable. Hopefully, you will be happy to see that there are fewer major points than before, and I feel optimistic that you can deal with these. Good luck, and hopefully I'll have a new version to read soon!

Major points

The study needs a clearer/stronger purpose. Overall, I found it, for the most part, clear to follow what you did, but it is still not that clear why you did it. Why are you looking at both crows and chickens? For example, is there a hypothesis about the evolution of inhibitory control /behavioural flexibility that you can test with a study using these two species? Do you expect crows or chickens to be better or worse at these things based on their ecology/social structure etc? In addition, you can also use this study to stress the importance of investigating cognitive abilities in species not necessarily considered clever (1) to challenge potentially incorrect assumptions and 2) if such species really are less clever, to broaden the range of cognitive ability investigated in cognitive studies – presumably a range of cognitive ability exists in nature and, if research is spread across that range, we will gain a better understanding of how different abilities evolve and why some species are better than others in certain abilities). Using tests that both species, at least physically, can do without difficulty (both can putt their head in tubes and remove lids to access food) allows a reasonably fair comparison.

Please mention that differences in how the crows and chickens were raised and tested (e.g. differences in social and physical environment, differences in reward value, novel vs familiar testing locations, difference in previous experience of transparent objects) could also potentially lead to/increase or mask/decrease differences in how these two species performed on cognitive tests, thus making comparison challenging. I understand that a lot of these things are hard to control for and that it is very challenging to ensure two species have the exact same experiences – especially if you want to have good welfare for your study animals! However, the fact remains that differences, such as described above, between the two study groups could have implications for your results and the honest thing to do is to acknowledge that.

Finally, please expand on potential causes of individual differences in the discussion. I understand if you cannot say what caused individual differences here, but I would like you to more than just touch upon explanations. You could discuss the main proposed explanations for individual differences in cognitive tests and maybe make some

suggestions for future research – this would tie in nicely with your intro in which you emphasize, and rightly so, the importance of individual differences in cognitive variation.

Minor points

Page 1, line 19: replace 'individuals' with study animals – as individual could refer to people carrying out the experiment.

Page 2, line 39: Claudia Wascher was (as the writing was done in the past)

NB: From this point onwards line numbers refer to continuous line numbering that starts on page 2

Line 9: move 'e-mail:' onto line 10 – then all the info about email contact on one line

Line 15: The word 'conflicting' could imply that this course of action is in some way negative – 'different' could work better here. Also, there should be a comma after 'In the present study'

Line 20: at this point it is not clear what 'acquisition phase' is, therefore rephrase to make clear what the chickens did, you can put acquisition phase in brackets afterwards so that you can then use this term later. Also, it is not clear what 'correct trial' is – perhaps you can remove the bits in brackets and just say that crows performed better.

Line 25: is learning task the reversal learning task? In general, I would stick to using 'reversal learning task' as there are lots of different learning tasks in cognition research

Line 27: In contrast,

Lines 30-31: Fantastic, individual differences are very important for evolution, so it is great you focus on these as well as between species differences, this makes me look forward to reading on.

Line 37: an animal's

Lines 38-39: 'or inhibit' I think would read better – it seems like there is a bit of an overlap between behavioural inhibition and behavioural flexibility as both involve inhibiting actions – I would like this to be discussed and the differences between them made clear. By doing a detour task, individuals may have strengthened their inhibitory control and so perform better on the reversal learning task than they would if they did not do the detour task (e.g. because they are better at inhibiting response to go to previously rewarded cue). Thus, you maybe have some order effects here that you should mention. I had similar comments on lines 46-48 so put this discussion there if you feel it fits better there.

Lines 39-42: This needs more detail, make it clear that they learn that one cue is rewarded and one not in the acquisition phase and then the cue-reward contingencies are switched so unrewarded becomes rewarded and vice versa in the reversal phase. Also remove the brackets inside the reference brackets

Line 56: Why do you list references for cognitive flexibility when you are talking about inhibitory control. Doing so makes sense if cognitive flexibility and inhibitory control are linked, if so, say how to justify including these refs – NB this possibly ties in with my comment on Lines 38-39

Lines 66-69: Group size effects on inhibitory control or that there is maybe a lack of link between inhibitory control and innovation is not relevant here so I would remove this discussion

Line 73: 'Individual differences ...are' – would the references you have for that inhibitory is investigated in lots of species fit better in this paragraph (I assume these studies compared between individuals?)

Lines 73-75: only recently has ...performance been acknowledged.

Lines 79: cognitive performance

Lines 81-92: Stick to just examples of inhibitory control and behavioural flexibility tests in corvids, that is what is relevant for this study

Lines 94-95: Please discuss this statement – why has research focused more on crows than fowl, why is it important to investigate fowl as well as crows. This can be a good opportunity to sell your study

Lines 94-104: included references that inhibitory is affected by housing/ranging. How did fowl perform on the reversal learning test in the Zidar study, e.g. what were the proportion of pass or fail?

Lines 102-104: so should we then expect similar cognitive abilities in both species? What could explain, for example, why fowl perform worse on inhibitory control tests than crows – maybe ecological/biological reason (maybe really do have lower inhibitory control) but could also be to do with how they are housed/raised? For example, if fowl kept in industrial, unnatural conditions, this could mean high stress...could this lead to poorer inhibitory control?

Lines 108-113: I like that you are not necessarily expecting crows to perform better than chickens just because we consider crows as 'brainy birds'. It is important to give all species/individuals the benefit of the doubt in terms of cognitive tests and not have prior assumptions about how well they will perform, so I appreciate that you do this. I feel this section could be added to – why are crows considered brainy birds and not chickens? Maybe because we do a lot more cognitive studies in crows (you see a bit of a spiral here perhaps, we do lots of cognitive studies in crows because we consider them smart and therefore interesting and then they do well so we find them more interesting and we do more studies...) – but fowl are demonstrating that they are cognitively able to...another interesting point is why do we consider chickens to not be brainy (without really doing research into chicken cognition) – this might have something to do with that, as they are livestock, we consider them more as 'commodities' than 'cognitive beings'

Line 118: Could it be an issue that you use one crow from another species? I would add a bit here to justify why you did that.

Lines 125-126: ejected from the nest

Line 129: the flock

Line 130: what were the dimensions for the space the chickens lived in indoors, and outdoors (or for the latter could they roam freely outside – even then would be nice to know their range)

Lines 149-150: 'sufficiently long that the birds needed to insert their head...to obtain the reward'

Line 153: remove 'considered'

Lines 159-161: how does this measure relate to inhibitory control?"

Lines 179-180: The chicken was tested though, she just didn't reach the criteria, so better to say that she wasn't included in further testing/analysis

Lines 182: you imply all birds developed a side bias, was this the case? If not say, 'if a subject...' also please describe the side bias correction procedure?

Lines 185-187: why did the 'rules' for rewarded trials on one side or the other differ between crows and chickens?

Line 200- what experiment?

Lines 204 -205: Do you get different result, qualitatively, if you remove these non-learners from the analysis? – if you do I suggest you report in a supplementary.

Make sure you keep the order in which tests are discussed the same in each section, i.e., always cylinder test and then reversal learning or vice versa

First paragraph of discussion should be a summary of results and I would also like a recap of why you did what you did (can come first, e.g. We here... to ...). You go off on a tangent in the first paragraph speaking about what might cause individual differences. Discussing this is definitely valid, but should be done in a separate paragraph

Lines 220 – 224: So, crows and chickens had similar performance in the reversal learning test – some chickens even, apparently, doing better than the crows! (at least one got the reversal stage in 30 trials compared to the quickest crow at 60 trials)

Lines 229-232: I like that you discuss the issue of selection biases – I would move things around a little though, as it currently sounds like you go back on yourself in terms of topics. Perhaps it works better to start a paragraph with 'Cognitive studies that use voluntary participation, as we did, may not sample the full range of personalities in a population, which can be problematic as personality can effect cognition (refs). Therefore, developing methods that ...is an important focus for future cognitive research. Nevertheless, despite using voluntary participation, we found considerable individual differences in our study sample ... then you can talk about potential causes of such differences – maybe in a following paragraph. Of course, you can rephrase this as you please, I only put it as a guideline that hopefully helps, and this paragraph may even work better elsewhere in the discussion (i.e. it doesn't need to come after the summary of results)

Line 233: withing should be within

Line 237: could say 'not traditionally considered' then does not imply that they don't have these skills (it's more that we stereotype them as stupid)

Line 238: Individual differences

Line 240: I appreciate that you admit to this, that small sample size is an issue in cognitive studies in general is important to discuss. Besides making statistics hard there is the issue that outliers may have a greater impact. Kudos to you for suggesting possible solutions – I like that you bring up collaboration as a solution, combining forces seems a good way forward!

Line 255- are considered

Lines 256 – 261: Is there evidence that crows can do these things and chickens can't? Or is there just a lack of tests for these cognitive skills in chickens. If the former, then I would expand this paragraph with discussion as to why crows may have these skills but not chickens, cognitive skills must provide a benefit to the species that have them as cognition is costly, so what benefit do crows get from these skills that chickens do not? If the latter then I would remove the 'compared to chickens' from line 257 and instead later say something along the lines of 'As chickens seem to show poor inhibitory control they could be expected to perform more poorly in tests of these cognitive skills, or possibly lack

these skills altogether. Further research is needed... I think this leads into the final sentence of this paragraph – which I think makes an important point, chickens can pass inhibitory control tests, which indicates that they do have inhibitory control and so could potentially also have these higher cognitive skills – maybe we need to think more carefully about how to design ‘chicken friendly’ tests – e.g. overcoming issues of motivation and perception

Line 268: in which redbilled...

Lines 271-272: Well said!

Lines 281-282: That does not necessarily imply that detour tasks don't measure inhibitory control though – it could be that inhibitory control is a complex factor that is made up of different independent aspects and different tests assess different aspects (this seems to be the case for impulsivity, a trait that is directly linked to inhibitory control – generally lower IC results in higher impulsivity) and so I would not be surprised if it was the case for IC as well. Of course, it is important to think about what else might be being measured/influencing results in inhibitory control tests and how we can control for this. I would also like discussion of the other inhibitory control tests and why they are better (if they are – or are they flawed also? Some factors, e.g. variation in motivation, perception, how easily distracted individuals are, stress, etc are issues for pretty much all cognitive tests)

Lines 283-286: How do you expect experience of transparency effected crows' performance compared to chickens – could this explain why crows did better than chickens

Lines 286-287: This sentence seems out of place – and also that there is information missing? Maybe have this in a separate paragraph e.g we found... other studies... why might a lack of correlation be present? Why might correlation be found in some species? Here you can discuss your finding that crows had this correlation but not chickens – as discussion of that seems to be lacking at the moment

Line 289: in our experiments,

General minor points:

You use ‘in order to’ a lot, have a look and see if these can just be replaced with ‘to’ so you are not using redundant words

Be consistent in what term you use to refer to something, e.g. inhibitory control, impulse control, motor impulse control... pick one and stick to it – otherwise it can get confusing. You can always mention other names for a term the first time you give it, e.g. inhibitory control (aka...)

You could work on concluding sentences for paragraphs in your intro and discussion, sometimes it can feel like a paragraph ends quite abruptly. Finishing a paragraph with a sentence that summarises the overall message of that paragraph, emphasizes what still needs to be done and, where possible segues into the next paragraph would make the article much more pleasant to read – it definitely helps with ‘flow’ and staying on topic.

It is possible that the opaque stage of the cylinder could be affected by how well individuals understand object permanence, an individual/species with better understanding of this may learn the detour faster.

Be careful about not going off on tangents and stick to the topic of your paper – eg inhibitory control and reversal learning, discussion of other cognitive tasks, for example, that do not relate to these things, while interesting, is not relevant.

Appendix B

ARU Cambridge
East Road
CB1 1PT
www.aru.ac.uk

Dear Dr Wascher

The Editors assigned to your paper RSOS-210504 "Learning and motor-impulse control in crows and domestic chickens" have now received comments from reviewers and would like you to revise the paper in accordance with the reviewer comments and any comments from the Editors. Please note this decision does not guarantee eventual acceptance.

Please submit your revised manuscript and required files (see below) no later than 21 days from today's (ie 02-Aug-2021) date. Note: the ScholarOne system will 'lock' if submission of the revision is attempted 21 or more days after the deadline. If you do not think you will be able to meet this deadline please contact the editorial office immediately.

on behalf of Professor Kevin Padian (Subject Editor)
openscience@royalsociety.org

Associate Editor Comments to Author:

Many thanks for your patience while we sought reviewers of the paper - this has been more difficult than we'd hoped, but the reviewers who have kindly reported on the paper recommend a number of changes that we'd like you to incorporate into the manuscript. If you need a short extension on the revision deadline, please let the editorial office know.

Thank you very much for your and the reviewer's input to our manuscript, we really appreciate this. We absolutely understand the challenges of these difficult times and that the review process might become delayed.

Reviewer comments to Author:

Reviewer: 1

Comments to the Author(s)

REVIEW FOR LEARNING AND MOTOR-IMPULSE CONTROL IN CROWS AND DOMESTIC
CHICKENS

Overview of study

This study aimed to investigate inhibitory control and behavioural flexibility in crows and domestic fowl. Inhibitory control was measured with a detour task and behavioural flexibility was measured in a reversal learning test. Crows performed better in the inhibitory control test than chickens and no difference was found in terms of reversal learning. The authors suggest that this implies crows are better at inhibitory control than chickens, but also point out that their small sample sizes make statistical comparison difficult. The study also touched upon within species differences/individual differences in cognition and repeatability of cognitive performance across different tests. I reviewed an earlier version of this manuscript and I was excited to see what it looked like now. I feel that, overall, it is much improved, and I am grateful that you seem to have taken my previous comments into consideration. I see promise in this paper to be interesting in itself and a good addition to the field of animal cognition –

though it is not quite there yet. I do have some points that I would like to see addressed before I would consider this study acceptable. Hopefully, you will be happy to see that there are fewer major points than before, and I feel optimistic that you can deal with these. Good luck, and hopefully I'll have a new version to read soon!

Thank you very much for your comment. Thank you for your previous, constructive feedback which helped us to improve the manuscript.

Major points

The study needs a clearer/stronger purpose. Overall, I found it, for the most part, clear to follow what you did, but it is still not that clear why you did it. Why are you looking at both crows and chickens? For example, is there a hypothesis about the evolution of inhibitory control /behavioural flexibility that you can test with a study using these two species? Do you expect crows or chickens to be better or worse at these things based on their ecology/social structure etc? In addition, you can also use this study to stress the importance of investigating cognitive abilities in species not necessarily considered clever (1) to challenge potentially incorrect assumptions and 2) if such species really are less clever, to broaden the range of cognitive ability investigated in cognitive studies – presumably a range of cognitive ability exists in nature and, if research is spread across that range, we will gain a better understanding of how different abilities evolve and why some species are better than others in certain abilities). Using tests that both species, at least physically, can do without difficulty (both can put their head in tubes and remove lids to access food) allows a reasonably fair comparison.

Thank you for the comment. We do not expect crows or chickens to do better in terms of ecology or social structure. Both species are living in complex groups and arguably probably share a similar feeding ecology. We have commented on the similar social structure in the introduction 'Both groups, corvids and chicken are considered to organise in complex social systems (Garnham & Løvlie, 2018; Wascher, 2018), which is assumed a driving factor in the evolution of cognitive skills (Dunbar, 1998).' We have included further discussion on differences in ecology, e.g., crows being a food-caching species, chickens not caching food.

We have further added: 'With our study we intend to add to existing data on performances of different species in standardized experimental paradigms, which should be relatively comparable to each other. Additionally, we intend to challenge the broad assumption of certain 'brainy' bird species to excel at cognitive performance over less 'brainy' species, without testing the latter. By applying two widely used experimental test paradigms which both species, at least physically, can perform without difficulty, i.e., removing a reward from a tube and lids to access food, allows a reasonably fair comparison of cognitive performance between these two species.' (Ins. 158-165). With this intention

we hope to more clearly outline the purpose of our study and we hope this is now sufficiently strongly explained.

Please mention that differences in how the crows and chickens were raised and tested (e.g. differences in social and physical environment, differences in reward value, novel vs familiar testing locations, difference in previous experience of transparent objects) could also potentially lead to/increase or mask/decrease differences in how these two species performed on cognitive tests, thus making comparison challenging. I understand that a lot of these things are hard to control for and that it is very challenging to ensure two species have the exact same experiences – especially if you want to have good welfare for your study animals! However, the fact remains that differences, such as described above, between the two study groups could have implications for your results and the honest thing to do is to acknowledge that.

Thank you, we have elaborated on this in the discussion.

‘Additionally, we must acknowledge potential differences in experience, for example regarding development and raising, social and physical environment, differences in experience with cognitive tests, and potentially transparent materials could account for differences in cognitive performance between species. In general, it is challenging to keep groups of animals under the same conditions and experiences might even vary between individuals of the same group.’ (Ins. 327-332)

Finally, please expand on potential causes of individual differences in the discussion. I understand if you cannot say what caused individual differences here, but I would like you to more than just touch upon explanations. You could discuss the main proposed explanations for individual differences in cognitive tests and maybe make some suggestions for future research – this would tie in nicely with your intro in which you emphasize, and rightly so, the importance of individual differences in cognitive variation.

We have elaborated on this paragraph.

‘They can be caused by differences in attention and motivation [42]. A recent study found individual differences in cognitive performance in Australian magpies, *Cracticus tibicen dorsalis*, to be linked to group size, with individuals in larger groups performing better in cognitive tasks, thus providing empirical support for the social intelligence hypothesis [43]. Ultimately, individual differences in cognitive performance can affect individuals’ survival and reproductive success [26,39,44]. In humans, different psychometric tests assessing different cognitive processes are well established, however similar tests have only been developed and validated in a small number of non-human animal species [45]. We suggest a better understanding of the causes and consequences of individual variation in

cognitive performance will enhance our understanding of the evolution of cognitive processes. ' (Ins. 307-316)

Minor points

Page 1, line 19: replace 'individuals' with study animals – as individual could refer to people carrying out the experiment.

We changed to ,subjects'. Please note that we had to significantly shorten the abstract as it is limited to 200 words now (it was 2000 characters in the initial submission).

Page 2, line 39: Claudia Wascher was (as the writing was done in the past) NB: From this point onwards line numbers refer to continuous line numbering that starts on page 2
Changed.

Line 9: move 'e-mail:' onto line 10 – then all the info about email contact on one line
Done.

Line 15: The word 'conflicting' could imply that this course of action is in some way negative – 'different' could work better here. Also, there should be a comma after 'In the present study'
Changed as suggested, thank you.

Line 20: at this point it is not clear what 'acquisition phase' is, therefore rephrase to make clear what the chickens did, you can put acquisition phase in brackets afterwards so that you can then use this term later. Also, it is not clear what 'correct trial' is – perhaps you can remove the bits in brackets and just say that crows performed better.

We added 'initial discrimination learning phase' to explain what the acquisition phase is, hopefully this is a bit more intuitive. As suggested, we removed the information in brackets.

Line 25: is learning task the reversal learning task? In general, I would stick to using 'reversal learning task' as there are lots of different learning tasks in cognition research.
Changed to 'reversal learning experiment' because this is the term previously used in the abstract.

Line 27: In contrast,
Thank you, changed.

Lines 30-31: Fantastic, individual differences are very important for evolution, so it is great you focus on these as well as between species differences, this makes me look forward to reading on.

Thank you.

Line 37: an animal's
Thank you changed.

Lines 38-39: 'or inhibit' I think would read better – it seems like there is a bit of an overlap between behavioural inhibition and behavioural flexibility as both involve inhibiting actions – I would like this to be discussed and the differences between them made clear. By doing a detour task, individuals may have strengthened their inhibitory control and so perform better on the reversal learning task than they would if they did not do the detour task (e.g. because they are better at inhibiting response to go to previously rewarded cue). Thus, you maybe have some order effects here that you should mention. I had similar comments on lines 46-48 so put this discussion there if you feel it fits better there.

Excellent point, thank you very much. We now discuss this.

'It should be noted that the reversal learning task also requires a certain level of behavioural inhibition [64]. In the chickens, we have conducted the cylinder task before the reversal learning experiment, thus previous experience with the motor-inhibition task could have affected performance of chickens in the reversal learning task.' (Ins. 387-391)

Lines 39-42: This needs more detail, make it clear that they learn that one cue is rewarded and one not in the acquisition phase and then the cue-reward contingencies are switched so unrewarded becomes rewarded and vice versa in the reversal phase. Also remove the brackets inside the reference brackets

We have elaborated on this.

Line 56: Why do you list references for cognitive flexibility when you are talking about inhibitory control. Doing so makes sense if cognitive flexibility and inhibitory control are linked, if so, say how to justify including these refs – NB this possibly ties in with my comment on Lines 38-39

We have removed the references to cognitive flexibility.

Lines 66-69: Group size effects on inhibitory control or that there is maybe a lack of link between inhibitory control and innovation is not relevant here so I would remove this discussion

Done.

Line 73: 'Individual differences ...are' – would the references you have for that inhibitory is investigated in lots of species fit better in this paragraph (I assume these studies compared between individuals?)

But usually studies only report performance on species level.

Lines 73-75: only recently has ...performance been acknowledged.

Changed accordingly.

Lines 79: cognitive performance

Changed.

Lines 81-92: Stick to just examples of inhibitory control and behavioural flexibility tests in corvids, that is what is relevant for this study

Done.

Lines 94-95: Please discuss this statement – why has research focused more on crows than fowl, why is it important to investigate fowl as well as crows. This can be a good opportunity to sell your study

See comment above, we have elaborated on this.

Lines 94-104: included references that inhibitory is affected by housing/ranging. How did fowl perform on the reversal learning test in the Zidar study, e.g. what were the proportion of pass or fail? We have added this information: ,In a reversal learning task, it took chicks 7–64 (mean \$\pm\$ SE, 24.02 \$\pm\$ 1.61) trials to learn to initially discriminate between two colours and adult female chickens 15–93 trials (mean \$\pm\$ SE, 46.32 \$\pm\$ 3.75). In the reversal phase of the same study, chicks learned the reversed reward contingency after 10–62 trials (mean \$\pm\$ SE, 32.98 \$\pm\$ 1.76) and adult female chickens learned this after 24–100 trials (mean \$\pm\$ SE, 65.22 \$\pm\$ 4.53).' (Ins. 133-138)

Lines 102-104: so should we then expect similar cognitive abilities in both species? What could explain, for example, why fowl perform worse on inhibitory control tests than crows – maybe ecological/biological reason (maybe really do have lower inhibitory control) but could also be to do with how they are housed/raised? For example, if fowl kept in industrial, unnatural conditions, this could mean high stress...could this lead to poorer inhibitory control?

Yes, we think we should. It might be specific propensities of the cylinder task causing lower performance in chickens. The task has previously been widely criticized, e.g., Horik JO van, Langley EJ, Whiteside MA, Laker PR, Beardsworth CE, Madden JR. 2018. Do detour tasks provide accurate assays

of inhibitory control? Proceedings of the Royal Society B: Biological Sciences.285(1875):20180150. We elaborate on this in the discussion.

Lines 108-113: I like that you are not necessarily expecting crows to perform better than chickens just because we consider crows as 'brainy birds'. It is important to give all species/individuals the benefit of the doubt in terms of cognitive tests and not have prior assumptions about how well they will perform, so I appreciate that you do this. I feel this section could be added to – why are crows considered brainy birds and not chickens? Maybe because we do a lot more cognitive studies in crows (you see a bit of a spiral here perhaps, we do lots of cognitive studies in crows because we consider them smart and therefore interesting and then they do well so we find them more interesting and we do more studies...) – but fowl are demonstrating that they are cognitively able to...another interesting point is why do we consider chickens to not be brainy (without really doing research into chicken cognition) – this might have something to do with that, as they are livestock, we consider them more as 'commodities' than 'cognitive beings'

We have elaborated on this in the introduction:

'In contrast to domestic chickens, crows are widely considered 'brainy' birds [28], thus we could expect high levels of motor inhibition in crows, but not in chickens. Recent research challenges this view and highlights the cognitive abilities of chickens [37]. Further, it could be expected that crows reach learning criterion in the reversal learning task quicker compared to chickens and to show higher behavioural flexibility. We do not necessarily share this expectation and aim to contribute to knowledge about individual performances in cognitive tasks in two well-established model species. With our study we intend to add to existing data on performances of different species in standardized experimental paradigms, which should be relatively comparable to each other. Additionally, we intend to challenge the broad assumption of certain 'brainy' bird species to excel at cognitive performance over less 'brainy' species, without testing the latter. By applying two widely used experimental test paradigms which both species, at least physically, can perform without difficulty, i.e., removing a reward from a tube and lids to access food, allows a reasonably fair comparison of cognitive performance between these two species.' (Ins. 145-165)

Line 118: Could it be an issue that you use one crow from another species? I would add a bit here to justify why you did that.

We consider carrion crows and hooded crows as one species (*Corvus corone*). The 'long' scientific name of hooded crows is (*Corvus corone cornix*) and carrion crows (*Corvus corone corone*). In Austria carrion crows and hooded crows also hybridize.

Lines 125-126: ejected from the nest
Thank you changed.

Line 129: the flock
Changed.

Line 130: what were the dimensions for the space the chickens lived in indoors, and outdoors (or for the latter could they roam freely outside – even then would be nice to know their range)
We have added 'The chickens outdoor ranging area was 334 m², including the indoor area they were kept in at night, which was 13 m².' (Ins. 182-183)

Lines 149-150: 'sufficiently long that the birds needed to insert their head...to obtain the reward'
Changed.

Line 153: remove 'considered'
Done.

Lines 159-161: how does this measure relate to inhibitory control?"
We added: , Individuals able to inhibit trying to reach the reward directly from the front but retrieving the reward from the long side of the cylinder, and thus being able to detour, are considered better at motor-inhibition.' (Ins. 217-219)

Lines 179-180: The chicken was tested though, she just didn't reach the criteria, so better to say that she wasn't included in further testing/analysis
We have rephrased to: '...was not included in further testing in the reversal phase of the experiment.'

Lines 182: you imply all birds developed a side bias, was this the case? If not say, 'if a subject...' also please describe the side bias correction procedure?
Thank you, changed to ,if' and added 'whereupon we reverted to the normal pseudo-randomized trial schedule and always rewarded the non-preferred side until that subject chose the non-preferred side once.' (Ins. 246-249)

Lines 185-187: why did the 'rules' for rewarded trials on one side or the other differ between crows and chickens?
We had no particular reason for this.

Line 200- what experiment?
Changed to ,reversal task'

Lines 204 -205: Do you get different result, qualitatively, if you remove these nonlearners from the analysis? – if you do I suggest you report in a supplementary. Make sure you keep the order in which tests are discussed the same in each section, i.e., always cylinder test and then reversal learning or vice versa. First paragraph of discussion should be a summary of results and I would also like a recap of why you did what you did (can come first, e.g. We here... to ...). You go off on a tangent in the first paragraph speaking about what might cause individual differences. Discussing this is definitely valid, but should be done in a separate paragraph

If removed chickens reach learning criterion between 30-130 trials, which we suggest is not a difference worth reporting. We presently do not present supplementary materials, however we do present the original data, therefore the interested reader can look this up. We have changed the first paragraph of the discussion as suggested.

Lines 220 – 224: So, crows and chickens had similar performance in the reversal learning test – some chickens even, apparently, doing better than the crows! (at least one got the reversal stage in 30 trials compared to the quickest crow at 60 trials)

Indeed, which illustrates even more how important it is to consider individual differences in performance more widely when investigating the cognitive abilities in animals.

Lines 229-232: I like that you discuss the issue of selection biases – I would move things around a little though, as it currently sounds like you go back on yourself in terms of topics. Perhaps it works better to start a paragraph with 'Cognitive studies that use voluntary participation, as we did, may not sample the full range of personalities in a population, which can be problematic as personality can effect cognition (refs). Therefore, developing methods that ...is an important focus for future cognitive research. Nevertheless, despite using voluntary participation, we found considerable individual differences in our study sample ... then you can talk about potential causes of such differences – maybe in a following paragraph. Of course, you can rephrase this as you please, I only put it as a guideline that hopefully helps, and this paragraph may even work better elsewhere in the discussion (i.e. it doesn't need to come after the summary of results)

Thank you for the suggestion. We have moved this paragraph and now discuss with other limitations of the study and we have also rephrased the paragraph.

Line 233: withing should be within
Changed.

Line 237: could say 'not traditionally considered' then does not imply that they don't have these skills (it's more that we stereotype them as stupid)

We have rephrased the sentence slightly different than suggested and hope this is now captured better.

Line 238: Individual differences

Done.

Line 240: I appreciate that you admit to this, that small sample size is an issue in cognitive studies in general is important to discuss. Besides making statistics hard there is the issue that outliers may have a greater impact. Kudos to you for suggesting possible solutions – I like that you bring up collaboration as a solution, combining forces seems a good way forward!

Thank you very much!

Line 255- are considered

Changed.

Lines 256 – 261: Is there evidence that crows can do these things and chickens can't? Or is there just a lack of tests for these cognitive skills in chickens. If the former, then I would expand this paragraph with discussion as to why crows may have these skills but not chickens, cognitive skills must provide a benefit to the species that have them as cognition is costly, so what benefit do crows get from these skills that chickens do not? If the latter then I would remove the 'compared to chickens' from line 257 and instead later say something along the lines of 'As chickens seem to show poor inhibitory control they could be expected to perform more poorly in tests of these cognitive skills, or possibly lack these skills altogether. Further research is needed... I think this leads into the final sentence of this paragraph – which I think makes an important point, chickens can pass inhibitory control tests, which indicates that they do have inhibitory control and so could potentially also have these higher cognitive skills – maybe we need to think more carefully about how to design 'chicken friendly' tests – e.g. overcoming issues of motivation and perception

We suggest food caching as a potential biological cause for better impulse control in crows compared to chickens. We have added a sentence on this in the discussion:

'Carrion crows are a food caching species, which regularly hide food for short periods of time [59]. The regular practice to leave food behind for later consumption could lead to better motor inhibitory control in crows compared to chickens, which are a non-caching species.' (Ins. 359-362)

Line 268: in which red billed...
Changed.

Lines 271-272: Well said!
Thank you very much.

Lines 281-282: That does not necessarily imply that detour tasks don't measure inhibitory control though – it could be that inhibitory control is a complex factor that is made up of different independent aspects and different tests assess different aspects (this seems to be the case for impulsivity, a trait that is directly linked to inhibitory control – generally lower IC results in higher impulsivity) and so I would not be surprised if it was the case for IC as well. Of course, it is important to think about what else might be being measured/influencing results in inhibitory control tests and how we can control for this. I would also like discussion of the other inhibitory control tests and why they are better (if they are – or are they flawed also? Some factors, e.g. variation in motivation, perception, how easily distracted individuals are, stress, etc are issues for pretty much all cognitive tests).

Agreed, we have removed the sentence referring to correlations between different measures of inhibitory control. We feel that discussion of other inhibitory control tests is beyond the scope of the present paper and this is discussed elsewhere (e.g. Beran, M. (2018). Self-control in animals and people. Academic Press).

Lines 283-286: How do you expect experience of transparency effected crows' performance compared to chickens – could this explain why crows did better than chickens.
Could be, we have added a sentence to discuss this.

'Experience with transparent materials could be a potential cause for crows performing better compared to chickens, however in great tits, Parus major, general experience with a transparent wall did not improve performance in the cylinder task [11].' (Ins. 402-405)

Lines 286-287: This sentence seems out of place – and also that there is information missing? Maybe have this in a separate paragraph e.g we found... other studies... why might a lack of correlation be present? Why might correlation be found in some species? Here you can discuss your finding that crows had this correlation but not chickens – as discussion of that seems to be lacking at the moment
We have added a discussion on this.

'We found a positive correlation between learning performance in the acquisition and reversal phase

in crows but not in chickens. This indicates a relationship between reward-based stimulus association learning and behavioural flexibility. Several studies evidence a lack of correlation in performance between the acquisition and reversal learning phase [67]. The described species difference between crows and chickens is interesting and warrants further investigation, however due to the low sample size of the present study should not be over-interpreted.’ (lns. 407-412)

Line 289: in our experiments,
Done.

General minor points:

You use ‘in order to’ a lot, have a look and see if these can just be replaced with ‘to’ so you are not using redundant words.
Done.

Be consistent in what term you use to refer to something, e.g. inhibitory control, impulse control, motor impulse control... pick one and stick to it – otherwise it can get confusing. You can always mention other names for a term the first time you give it, e.g. inhibitory control (aka...)
We now use the term ,motor inhibitory control’ throughout.

You could work on concluding sentences for paragraphs in your intro and discussion, sometimes it can feel like a paragraph ends quite abruptly. Finishing a paragraph with a sentence that summarises the overall message of that paragraph, emphasizes what still needs to be done and, where possible segues into the next paragraph would make the article much more pleasant to read – it definitely helps with ‘flow’ and staying on topic.
We have added a summary sentence to some of the paragraphs and further rephrasing hopefully improved the flow of the manuscript.

It is possible that the opaque stage of the cylinder could be affected by how well individuals understand object permanence, an individual/species with better understanding of this may learn the detour faster.
This is actually a very good point, thank you very much. We incorporated discussion on this.

Be careful about not going off on tangents and stick to the topic of your paper – eg inhibitory control and reversal learning, discussion of other cognitive tasks, for example, that do not relate to these things, while interesting, is not relevant.

We have removed references to other cognitive tasks.

Reviewer: 2

Comments to the Author(s)

This study by Wascher, Allen and Szapl shows how crows and domestic chickens differ in their motor inhibition skills, but not necessarily in overall behavioural flexibility.

To this end, the authors provided a limited sample of individuals of both species with a) a single reversal-learning task investigating behavioural flexibility and b) a cylinder task to test for inhibitory control.

Finally, the authors highlight the importance of individual variation in performance in both species.

I have two main content remarks regarding this study (and try to provide suggestions on how to address these):

I was surprised to see that the authors did not elaborate more on the conceptual overlaps and implications of choosing these two tasks:

Reversal learning (and especially the reversal phase) has been suggested in older literature to comprise of the ability to inhibit responses to stimuli with a previous positive valence (being rewarded) and the ability to shift the attention to previously unrewarded stimuli (e.g. Dias et al., 1996; Lai et al., 1995; Macphail, 1971, 1976).

However, more recent studies (Powers 1989; Bonte, Flemming & Fagot 2011; Fagot, Bonte, & Hopkins 2013; Bonte, Kemp & Fagot 2014) agree that reversal learning rather measures the ability to shift the attention (as a measure for flexibility), than the inhibition of responding to a previously rewarded stimuli. It seems that what the authors find is along these lines of this newer literature, as they did not find a correlation between the reversal task and the inhibition task.

Thank you very much for the comment. We have now added this in the introduction.

‘In the past, the reversal learning task, next to a test of behavioural flexibility, also has been considered a test of behavioural inhibition, as individuals have to inhibit a previously learned behaviour to succeed when the reward contingency changes [17–19]. However, based on tests in humans, primates and rodents, this view has recently been contested [3,20–23].’ (Ins. 95-98)

Nevertheless, it might be telling to compare delta values in the reversal task (trials required in the reversal subtracted by trials in the acquisition). If there is a correlation between pure inhibition (as in the cylinder task) and such values (which would account for general differences in learning) would indicate that inhibition plays a more prominent role in the reversal phase. Whereas, if there is still no correlation between tasks, it would strengthen the argument that reversal learning instead measures the ability to shift attention.

However, I understand this analysis might be challenging given the low sample size of individuals performing in all stages (acquisition and reversal) and both tasks (reversal learning and cylinder task). Thank you very much for the suggestion. We have calculated delta values and correlated with performance in cylinder task, however, there is no significant correlation (neither overall (crows and chickens together) nor in the chickens only). We have included the latter analysis in the results section.

As the authors highlight the importance of considerable individual variation in performance, I would have been interested in an analysis and statement as to what factors (sex, age, personalities?) might influence this variation. Possibly a model on the primary principle component of the measures could inform if any specific factor would affect performance?

This is an interesting suggestion. Unfortunately, with our present dataset this analysis does not seem sensible to us, e.g., we have tested a limited number of male individuals in crows, only females in chickens. Similarly, there chickens are all approximately the same age. Unfortunately, we have not assessed personality in the tested individuals.

Below are some minor stylistic comments:

Line 13: should read "the ability of an animal to alter its behaviour" or "the ability of animals to alter their behaviour"

Thank you, changed.

Lines 19-20: It would be clearer just to state how many individuals complete the entire reversal-learning task

Done.

Line 37: should read "an animal's ability to alter its behaviour"

Done.

Line 90: remove the opening parenthesis before Kabadayi et al.
Done.

Line 129: missing "The" before flock
Thank you, changed.

Line 233: should read: "...pronounced individual differences within..."
Done.

Line 234: check the spelling of "focuses"
We think we spelled it correctly.

Line 238: should read: "individual differences might exist..."
Done.

Line 254: should read: "...crows compared to chickens."
Done.

Line 257: check the spelling of "suppress"
Changed.

Line 275: should read: "...the question of which cognitive abilities..."
Done.

Please also check the correct use of commas in several places.
Done.

Appendix C

Dear Authors,

I was happy to have the opportunity to review this manuscript again. I feel that you improved it a lot since the last version, and I was satisfied with how you dealt with my previous comments. I don't have any major comments on it anymore. Below are some minor comments, just things I noticed, or thought of, as I read it through that I think would improve the final version. Line numbers refer to the version of the manuscript with changes not showing (i.e. first version in the proof file).

All the best!

Line 13: include 'the' before performance

Lines 19-20: did performance in the two tasks correlate in crows?

Line 31: the survival

Line 32-33: I think this reads better if you replace 'and to' with a comma

Line 39: take out 'conflicting'

Line 41: should be 'learnt' not 'learned' if you are writing in UK English – you make the same error later on as well

Line 39-60: I think either use 'behavioural inhibition' or '(motor) inhibitory control' it is a bit confusing perhaps to have two terms for the same thing, the latter I think works better as that is in your title

Line 63: individual differences

Line 67: the temporal

Line 70: should this be moved down one line to start a new paragraph?

Line 78:) is missing

Line 81: would be nice to point out that thus, chickens have been given less chances to demonstrate their cognitive abilities

Line 81-82: please say how affected by housing conditions

Lines 86-91: Do you have any info on number of trials needed to reach reversal criterion for any studies on corvids (like you have for junglefowl), it would be interesting to have that for comparison

Line 92-94: I think this final sentence would go better at the start of this paragraph, slightly rephrased e.g. 'The social system an animal belongs to is thought to be a key driver of their cognitive skills, therefore species that have similar social systems to crows could be expected to have similar cognitive skills to them. Chickens, like crows, organise in complex social systems, however few studies have investigated their cognitive abilities' – I think this works as a good way to segue between crows and chickens and justify why you look at both species

Lines 99- 104: I think this section can be shortened:

'...widely considered 'brainy' birds [28], thus we could expect high levels of motor inhibition, and faster reversal learning, in crows, compared to chickens. However, as recent research challenges this view and highlights the cognitive abilities of chickens [37], we do not necessarily share this

expectation. Instead, we aim to contribute to knowledge about individual performances in cognitive tasks in two well-established model species.'

Lines 103-104: You could mention here something about that you could expect them to perform similarly based on that both live in complex societies (ties in to what you said earlier)

Line 105: the performances, also 'standardised' if UK English

Line 107-108: I suggest this change in the text '...over a species that is stereotypically considered to be 'less' brainy', ...'

Line 108: remove 'By'

Line 129: I think it reads better to replace 'one rooster' with 'one male' otherwise refer to the females as hens.

Line 130: Of this flock, 10... also don't need to say 'domestic' here

Lines 131-132: Of these 10, 1 individual was later...

Lines 146-154: I think some information lacking in this section, why did the experimenter place the reward in an opaque tube – it's so birds can learn the detour/how to get rewards out of tube, right? Once they pass you assume they know how to retrieve reward from tube, I think you should mention these things, also mention earlier that opaque tube stage is the first stage of the detour test.

Line 151: I don't think you need the comma after was

Line 155: a transparent cylinder (same dimensions as opaque?)

Line 157: say why chickens did it in two goes, e.g. because they had lower motivation

Line 171: 'training individuals to comfortably remove' is slightly odd wording (could you train them to uncomfortably do so?) – could remove 'comfortably' or say that training was done until they were comfortable with this action (did not hesitate to remove lid when presented with feeder?)

Line 178: needs comma after 'Once a subject met criterion'

Lines 186-189: would be nice to have a reason why the difference in testing between crows and chickens here

Line 198: what was the purpose of calculating delta values? Would be helpful to explain here for non statistically minded people (like myself)

Line 214: I think better to say that species did not differ significantly

Line 230: could be helpful to recap that cylinder task is inhibition test

Line 231-234: this seems repetitive as already said in the results, I think here it would be enough to say that crows and chickens performed similarly in both stages of the reversal test.

Line 237: the focus

240: they can also be caused by

244: I would be careful about saying this provides support for social intelligence hypothesis, this links social complexity to cognitive ability and the idea that larger groups are more socially complex has been questioned, could just be better to say that social group size can affect cognitive performance

247 – 249: I would include something along the lines of: ‘Determining which factors underlie cognitive variation, how cognitive variation affects individual outcomes and developing and validating tests of cognitive processes for multiple species could help to...’ then a final sentence ‘This, in turn, will enhance...’

252: ,but also within,

256: ,connected to this,

258: missing)

259: I don’t think you need ‘as such’ also ‘in the future’

260: I would discuss differences in experience in a separate paragraph ‘We must also acknowledge that our study animals may have differed in experience, which could have influenced their performance in our cognitive tests. For example...’

262: ,potentially, (otherwise can sound like the material is potentially transparent)

264: I’d mention that even when animals are kept in the same conditions, experiences might vary

267: in the case of our chickens

272: ...affecting the behaviour of study animals and thus who participates in voluntary cognitive testing

272-273: It seems a little odd to talk about individual differences here, when there was a paragraph on individual differences earlier and this paragraph is on selection bias – you do already mention that personality can cause individual difs in cognitive tests earlier, so I don’t think you need to bring up indiv difs again here

274: I would leave out STRANGEness, I don’t think having this in adds anything and it seems like jargon

275: on our results

276: Individual differences

277-281: I think this deserves to be a paragraph in its own right, I think the focusing only on a small number of species considered as cognitively complex is a major downfall/criticism of comparative cognition and that you find a species not typically considered to be cognitively complex to be able to perform as well as a species considered to be cognitively complex in a cognitive task is one of the most important findings of your study. At the moment, it feels like a little addition on the end of a paragraph, and I think you should put more emphasis on it.

285-287: The bit on humans seems tangential/unnecessary

288: more selectively attended...than chickens. Therefore, crows could be expected to perform... than chickens

291-292: The idea that crows are better at inhibitory control because they are a caching species, could if you want go in the intro – it’s an ecological reason why you might predict a difference between these two species in this test (it’s up to you, it works fine here the discussion).

296: this cognitive ability

313: we conducted

317-319: I find this sentence a little oddly worded, perhaps 'Next to individual variation in cognitive performance, another factor to consider when comparing cognition between species is the question of which cognitive abilities different tests assess' could work better

332: relationship in crows but not in chickens

337-334: I think you make a valid conclusion however I feel you could add a little more – e.g. encourage future research in animal cognition to move away from focusing on animals that are already presumed smart and instead broaden the range of species investigated – this will help us better understand the role cognition plays for different species and how different cognitive traits evolve.

528: was reached

530: had not (otherwise sounds like experiment is ongoing, same reason for change suggested above)

Appendix D

ARU Cambridge
East Road
CB1 1PT
www.aru.ac.uk

Dear Dr Wascher

On behalf of the Editors, we are pleased to inform you that your Manuscript RSOS-210504.R1 "Learning and motor-impulse control in crows and domestic chickens" has been accepted for publication in Royal Society Open Science subject to minor revision in accordance with the referees' reports. Please find the referees' comments along with any feedback from the Editors below my signature.

Please submit your revised manuscript and required files (see below) no later than 7 days from today's (ie 20-Sep-2021) date. Note: the ScholarOne system will 'lock' if submission of the revision is attempted 7 or more days after the deadline. If you do not think you will be able to meet this deadline please contact the editorial office immediately.

on behalf of Kevin Padian (Subject Editor)
openscience@royalsociety.org

Associate Editor Comments to Author:
Comments to the Author:

The reviewers have suggested a few minor modifications or clarifications, but the work is well on the way to being ready for acceptance - congratulations and we'll look forward to receiving a final version that incorporates these modifications.

Reviewer comments to Author:
Reviewer: 1

Dear Authors,

I was happy to have the opportunity to review this manuscript again. I feel that you improved it a lot since the last version, and I was satisfied with how you dealt with my previous comments. I don't have any major comments on it anymore. Below are some minor comments, just things I noticed, or though of, as I read it through that I think would improve the final version. Line numbers refer to the version of the manuscript with changes not showing (i.e. first version in the proof file).

All the best!

Line 13: include 'the' before performance
Changed.

Lines 19-20: did performance in the two tasks correlate in crows?
We did not perform this correlation as we only tested four crows in the cylinder task.

Line 31: the survival
Changed.

Line 32-33: I think this reads better if you replace 'and to' with a comma
Changed.

Line 39: take out 'conflicting'
Done.

Line 41: should be 'learnt' not 'learned' if you are writing in UK English – you make the same error later on as well
To my knowledge both forms are correct for British English.

Line 39-60: I think either use 'behavioural inhibition' or '(motor) inhibitory control' it is a bit confusing perhaps to have two terms for the same thing, the latter I think works better as that is in your title
Done.

Line 63: individual differences
Done.

Line 67: the temporal
Changed.

Line 70: should this be moved down one line to start a new paragraph?
Done.

Line 78:) is missing
Changed.

Line 81: would be nice to point out that thus, chickens have been given less chances to demonstrate their cognitive abilities
We have added this.

Line 81-82: please say how affected by housing conditions
We have removed this sentence as we explain this further down in the text.

Lines 86-91: Do you have any info on number of trials needed to reach reversal criterion for any studies on corvids (like you have for junglefowl), it would be interesting to have that for comparison
We do not have this information available.

Line 92-94: I think this final sentence would go better at the start of this paragraph, slightly rephrased e.g. 'The social system an animal belongs to is thought to be a key driver of their cognitive skills, therefore species that have similar social systems to crows could be expected to have similar cognitive skills to them. Chickens, like crows, organise in complex social systems, however few studies have investigated their cognitive abilities' – I think this works as a good way to segue between crows and chickens and justify why you look at both species

We changed the sentence as suggested.

Lines 99- 104: I think this section can be shortened: '...widely considered 'brainy' birds [28], thus we could expect high levels of motor inhibition, and faster reversal learning, in crows, compared to chickens. However, as recent research challenges this view and highlights the cognitive abilities of chickens [37], we do not necessarily share this expectation. Instead, we aim to contribute to knowledge about individual performances in cognitive tasks in two well-established model species.'

Changed as suggested.

Lines 103-104: You could mention here something about that you could expect them to perform similarly based on that both live in complex societies (ties in to what you said earlier)

This section was changed.

Line 105: the performances, also 'standardised' if UK English

Changed.

Line 107-108: I suggest this change in the text '...over a species that is stereotypically considered to be 'less' brainy', ...'

This section was changed.

Line 108: remove 'By'

Done.

Line 129: I think it reads better to replace 'one rooster' with 'one male' otherwise refer to the females as hens.

Done.

Line 130: Of this flock, 10... also don't need to say 'domestic' here

Done.

Lines 131-132: Of these 10, 1 individual was later...

Done.

Lines 146-154: I think some information lacking in this section, why did the experimenter place the reward in an opaque tube – it's so birds can learn the detour/how to get rewards out of tube, right? Once they pass you assume they know how to retrieve reward from tube, I think you should mention these things, also mention earlier that opaque tube stage is the first stage of the detour test.

We have added this information.

Line 151: I don't think you need the comma after was

We removed the coma.

Line 155: a transparent cylinder (same dimensions as opaque?)

Yes, we have added this information.

Line 157: say why chickens did it in two goes, e.g. because they had lower motivation

We added this information.

Line 171: 'training individuals to comfortably remove' is slightly odd wording (could you train them to uncomfortably do so?) – could remove 'comfortably' or say that training was done until they were comfortable with this action (did not hesitate to remove lid when presented with feeder?)

We removed 'comfortably'.

Line 178: needs comma after 'Once a subject met criterion'

We added the coma.

Lines 186-189: would be nice to have a reason why the difference in testing between crows and chickens here

We had no reason for this.

Line 198: what was the purpose of calculating delta values? Would be helpful to explain here for non statistically minded people (like myself)

We have added this information.

Line 214: I think better to say that species did not differ significantly

Added.

Line 230: could be helpful to recap that cylinder task is inhibition test
We have added this.

Line 231-234: this seems repetitive as already said in the results, I think here it would be enough to say that crows and chickens performed similarly in both stages of the reversal test.
We removed this.

Line 237: the focus
Done.

240: they can also be caused by
Changed.

244: I would be careful about saying this provides support for social intelligence hypothesis, this links social complexity to cognitive ability and the idea that larger groups are more socially complex has been questioned, could just be better to say that social group size can affect cognitive performance
We removed this statement.

247 – 249: I would include something along the lines of: ‘Determining which factors underlie cognitive variation, how cognitive variation affects individual outcomes and developing and validating tests of cognitive processes for multiple species could help to...’ then a final sentence ‘This, in turn, will enhance...’
Changed as suggested.

252: ,but also within,
Changed.

256: ,connected to this,
Changed.

258: missing)
Changed.

259: I don’t think you need ‘as such’ also ‘in the future’
Changed.

260: I would discuss differences in experience in a separate paragraph 'We must also acknowledge that our study animals may have differed in experience, which could have influenced their performance in our cognitive tests. For example...'

Changed as suggested.

262: ,potentially, (otherwise can sound like the material is potentially transparent)

Changed.

264: I'd mention that even when animals are kept in the same conditions, experiences might vary

We have already mentioned this.

267: in the case of our chickens

Changed.

272: ...affecting the behaviour of study animals and thus who participates in voluntary cognitive testing

Changed as suggested.

272-273: It seems a little odd to talk about individual differences here, when there was a paragraph on individual differences earlier and this paragraph is on selection bias – you do already mention that personality can cause individual difs in cognitive tests earlier, so I don't think you need to bring up indiv difs again here

We removed the repetitive part of the sentence.

274: I would leave out STRANGEness, I don't think having this in adds anything and it seems like jargon

We feel it is important to refer to this new framework.

275: on our results

Changed.

276: Individual differences

Changed.

277-281: I think this deserves to be a paragraph in its own right, I think the focusing only on a small number of species considered as cognitively complex is a major downfall/criticism of comparative cognition and that you find a species not typically considered to be cognitively complex to be able to

perform as well as a species considered to be cognitively complex in a cognitive task is one of the most important findings of your study. At the moment, it feels like a little addition on the end of a paragraph, and I think you should put more emphasis on it.

Changed as suggested.

285-287: The bit on humans seems tangential/unnecessary

We removed this sentence.

288: more selectively attended...than chickens. Therefore, crows could be expected to perform... than chickens

Changed as suggested.

291-292: The idea that crows are better at inhibitory control because they are a caching species, could if you want go in the intro – it's an ecological reason why you might predict a difference between these two species in this test (it's up to you, it works fine here the discussion).

Thank you for the suggestion, we actually slightly prefer to have this aspect in the discussion.

296: this cognitive ability

Changed.

313: we conducted

Changed.

317-319: I find this sentence a little oddly worded, perhaps 'Next to individual variation in cognitive performance, another factor to consider when comparing cognition between species is the question of which cognitive abilities different tests assess' could work better

Changed as suggested.

332: relationship in crows but not in chickens

We prefer not to add this, as this would be repetitive to the previous sentence.

337-334: I think you make a valid conclusion however I feel you could add a little more – e.g. encourage future research in animal cognition to move away from focusing on animals that are already presumed smart and instead broaden the range of species investigated – this will help us better understand the role cognition plays for different species and how different cognitive traits evolve.

Thank you for the suggestion, we have elaborated on this in the conclusion.

528: was reached
Changed.

530: had not (otherwise sounds like experiment is ongoing, same reason for change suggested above
Changed.

Reviewer: 2

Comments to the Author(s)

The revised manuscript by Wascher, Allen and Szpl: 'Learning and motor inhibitory control in crows and domestic chicken' has largely improved from the previous version I reviewed.

I'm happy to see how diligently the authors have taken all comments into consideration.

Based on the small sample size the generalisability of the results is of course limited. However, the authors fully acknowledge this and highlight the importance of individual factors contributing to the results. In this sense it adds to a body of literature and a current trend to consider a subjects 'STRANGENESS' when investigating behavioural/cognitive functions.

I concur that ultimately, one forte of this study will be an additive contribution to a larger scale comparison with regard to individual differences. To this end I think it would be beneficial to provide as many details as possible for all subjects tested.

I have seen that age was listed in the ESM, but I wonder if it was possible to also provide information for each individual on other factors that are discussed to influence performance (e.g. housing conditions, rearing conditions, experimental experience, etc.) within the data table.

We have added information on age and previous experience.